# Recent Progress in Single and Combined Porosity-Evaluation Techniques for Porous Materials

**DOI:** 10.3390/ma15092981

**Published:** 2022-04-20

**Authors:** Yuqing Wang, Bo Zhou

**Affiliations:** Jiangsu Key Laboratory of Micro and Nano Heat Fluid Flow Technology and Energy Application, School of Environmental Science and Engineering, Suzhou University of Science and Technology, Suzhou 215009, China; 1911024003@post.usts.edu.cn

**Keywords:** porosity measurement, SANS, CT, SEM, NMR, MIP, WIP, adsorption, gas expansion

## Abstract

The accurate determination of the porosity and specific surface area of porous materials such as shale and cement plays a key role in gas-energy-storage estimation and exploitation, building-heat and humidity-transfer investigation, and permeability-characteristics evaluation. Therefore, it is crucial to select appropriate measurement methods to accurately study the porosity, as well as other properties, of porous materials. In this review, various porosity-measurement methods are discussed. The most recent research findings and progress in combined methodologies are introduced and summarized. The measurement medium and chemical composition of the sample affect the porosity-measurement results. Therefore, depending on the measurement properties of different methods and the characteristics of the sample, an appropriate method can be selected. Furthermore, various methods can be combined to obtain more accurate measurement results than individual methods.

## 1. Introduction

Over the past decades, methane, hydrogen, and other gaseous energy carriers have been extensively developed and widely utilized. Consequently, porous materials have attracted significant research interest for gas storage and transport. Natural media and artificial porous materials are essential for gas storage. Natural media such as coal and shale are primary energy sources [1]. Shale gas predominantly exists in adsorbed phases in organic matter and inorganic minerals and in free phases in fractures and intergranular pores [2,3]. Traditional measurement methods, such as mercury intrusion porosimetry (MIP), are typically used to characterize porosity [4,5]. Understanding the pore structure of shale is important for evaluating the storage of free gas. The porosity of artificial porous materials affects hydrate formation and their gas-storage capacity [6]. Moreover, artificial porous materials are suitable for high-pressure gas storage because of their large specific surface area (SSA). Metal-organic frameworks (MOFs), covalent organic frameworks (COFs), and porous aromatic frameworks (PAFs) have been extensively examined as CO_2_, CH_4_, and H_2_ gas-storage materials owing to their large pore volumes and surface areas [7,8,9]. The structures of these nanoporous materials must be fully described to elucidate the relationship between good structural qualities and gas-storage properties [10]. These artificial materials are typically characterized via adsorption analysis. In both natural and artificial porous materials, gases are stored as adsorbed and free gases. Quantitative porosity measurements of these materials help evaluate their free-gas content and gas-storage capacity.

In addition to fluid-storage capacity, the porosity of a material also significantly affects its transport behavior, including fluid flow, component diffusion, and heat transfer. Porous catalysts contain numerous ion/mass-absorption sites, abundant surface-reaction active sites, and sufficient mass-charge transfer channels, significantly increasing the ion-transport rate [11]. Accurate characterization of the SSA and porosity of materials is essential to evaluate their catalytic effects and improve chemical-reaction rates. Structured porous metals (e.g., Ni, Cu, Zn, and their oxides) have been used as electrodes in lithium-ion batteries or fuel cells owing to their advantages such as high conductivity, high porosity/SSA, controllable structure, and light weight, which significantly impact the porous-medium flow performance by affecting their fluid distribution and transport, hydrothermal properties, and electrical resistance. Furthermore, cementitious materials are typical porous structures employed in the field of architecture. The heat and moisture transfer of the building envelope are very complex processes that affect the buildings’ energy consumption and durability, as well as the thermal comfort of residents [12,13]. The pore structure of cementitious materials determines their macrophysical properties (e.g., permeability) and affects their durability [14,15,16,17]; hence, it is crucial to study the porosity of cementitious materials.

The selection of appropriate experimental methods is an important factor for low-porosity and low-permeability materials. Porosity-measurement methods are classified into radiation-detection and fluid-intrusion methods. Most fluid-intrusion methods, such as MIP, water-immersion porosimetry (WIP), adsorption analysis, and gas-expansion/immersion methods, have been widely used in previous porosity-measurement studies [5,18,19,20]. Recently, several radiation-detection methods have been introduced in the field of pore-structure characterization, including low-field nuclear magnetic resonance (LF-NMR), small-angle scattering technology, electron microscopy, and computed tomography (CT) scanning [21,22,23,24]. Although the application of various measurement methods has become well-established, and their measurement accuracy has been recognized to a certain extent, the following challenges still exist: (i) for fluid-intrusion methods, the molecular size of the measuring medium determines the diameter range of the open pores (connected to the outer surface) that can be penetrated; therefore, some tiny pores will be excluded; (ii) adsorption affects the gas-filling amount, resulting in inaccurate skeleton volume measurements; (iii) for CT and electron microscopy, high resolution limits the field of view and leads to a time-consuming image-processing step. Several scholars have combined methods to compare and verify the measurement results of different methods, increasing their credibility [24,25,26,27].

In this review, currently used porosity-measurement methods are classified into radiation-detection and fluid-intrusion-measurement methods. Each of these is briefly introduced, and their characteristics and measurement accuracies are compared. The implementation and conclusions of a combination of multiple methods are introduced. The advantages of the LF-NMR combined with other methods are discussed and the applicability of various measurement methods is clarified. In general, the present review provides resources for new researchers in related fields to gain insight into existing experimental methods and help them select suitable experimental methods.

## 2. Radiation-Detection Methods

Radiation-detection methods for porous media are based on the refraction, transmission, or scattering of rays by a solid skeleton. Among the various radiation-detection methods, electron microscopy uses electron beams, CT scanning utilizes X-rays, small-angle scattering uses X-rays or neutron rays, and NMR is based on the excitation of hydrogen atoms via electromagnetic radiation.

### 2.1. Electron Microscopy

Transmission electron microscopy (TEM) was designed according to the principle of optical projection microscopy. The samples must be thin enough (thickness of 100–500 nm) to be transparent to the electron beam. At present, the resolution of a 400–kV TEM instrument can reach 0.2 nm [21]. Furthermore, TEM should be used in conjunction with other methods when characterizing organic and carbon samples [28,29].

Scanning electron microscopy (SEM) is used to produce amplified images by replacing light waves with electrons; it can scan the sample surface with a resolution below 1 nm and an amplification of more than 4 × 10^6^ times [21]. Compared to traditional optical microscopy, SEM has a higher resolution, dynamic amplification range, and depth of field. It can also better analyze samples in detail when connected to an X-ray analyzer [30]. Figure 1 shows the SEM morphology slice-and-view process of cement-paste samples. The cement-paste samples were prepared with Type I Portland cement and deionized water with a water-to-binder ratio of 0.45. Then, the cement paste was cut using a low-speed diamond saw into 10 × 10 × 2 mm^3^ samples. 1000 slices of 2D SEM images were aligned and stacked into a 3D bounding box with dimensions 10 × 10 × 20 μm. More details can be found in Lim et al [31].

TEM and SEM can directly observe the pore morphology and identify pore types [32]. Figure 2 shows the schematic of TEM and SEM optical designs. They have been proven to be effective methods for characterizing the pore morphology and structure as follows: (i) Although TEM can probe nanoscale features, it requires thin samples that are transparent to the electron beam. Preparing a thin sample requires destructing the bulk sample and polishing the surface, which may cause artificial cracks and pores, in turn resulting in inaccurate imaging results [33,34]. Currently, many laboratories use argon ion-milling powder to complete the polishing process and maintain the true mineral texture and pore structure [35]. (ii) Owing to the high magnification involved in SEM, it can only provide a local pore morphology with a narrow area [27,36]. To overcome this drawback, multiple images can be spliced or reconstructed in 3D. However, the reconstruction process is complicated and time-consuming [37]. (iii) Sufficient image segmentation is required when using focused ion-beam (FIB)-SEM tomography to quantitatively analyze 3D structures. The segmentation method of choice is related to the accuracy of the quantitative porosity analysis. Čalkovský et al. compared the signal-processing algorithm, Otsu’s method, Darwinian particle swarm optimization (DPSO), harmony search optimization (HSO), and fuzzy c-means threshold algorithms, and found that all these algorithms slightly underestimated the true pore size. The derived criteria for selecting the intensity threshold at the pore–polymer interface were proposed for accuracy improvement [38].

### 2.2. Small-Angle Scattering Technology

Small-angle scattering technology is based on quantitatively interpreting the microstructure and porosity of samples using the relationship between the scattering radiation intensity and scattering angle obtained via neutron or X-ray irradiation. Two basic approaches are currently being adopted in the research of small-angle scattering technology: small-angle X-ray scattering (SAXS) and small-angle/ultra-small angle neutron scattering (SANS/USANS) [22]. Figure 3 shows the steps involved in SANS data analysis.

SANS/USANS can characterize nanopore structures and their confined fluid behavior. Owing to their high permeability of neutrons, SANS/USANS can detect the interior of samples and provide information on closed pores and pore-size structures of 1 nm–10 μm compared with X-rays [33,39]. Neutron scattering is more sensitive to the position of hydrogen and its isotopes, making it suitable for studying the contents of hydrogen and other hydrogen elements (solid and liquid).

Figure 4 shows the scattering profiles for shale samples named QD_1_-L3, QD_1_-L4, WX_2_-8, WX_2_-33, WX_2_-49, and WX_2_-54, from which the pore-size distribution (PSD) can be obtained. In Figure 4a, *Q* and *I*(*Q*) can be respectively defined as:(1)$$Q=\frac{4\pi }{\lambda }\sin\theta$$
(2)$$I(Q)=N{\left(\Delta \rho^{*}\right)}^{2}{\int V}^{2}(r)f(r)P(Q,r)dr$$
where *λ* is the neutron wavelength; *θ* is the Bragg angle, which is the half of the scattering angle; *N* is the pore number density; (Δ*ρ*^*^)^2^ is the scattering contrast, which is equal to (*ρ*_s_^*^ − *ρ*_p_^*^)^2^, that is, the square of the difference between the scattering length density (SLD) of the matrix and that of the pores (generally taken to be zero); *V*(*r*) is the spherical volume; *f*(*r*) is the PSD; *r* is the spherical pore radius; and *P*(*Q*, *r*) is the spherical form factor. Here, the pore size can be estimated by Bragg’s law with *Q* as *r* = π/*Q* in radius or *d* = 2π/*Q* in diameter. The approximately linear relationship between log(*I*/(*Q*)) and log(*Q*) in Figure 4a indicates that the six samples tested have a fractal pore structure. Figure 4b shows that the PSD has a peak at approximately 2 nm for each shale sample, which indicates either the existence of inaccessible pores or heterogeneity at a pore size of approximately 2 nm for the samples tested.

Several challenges still exist that limit the application of small-angle scattering technology, as follows: (i) Scattering techniques, including SAXS and SANS, allow the characterization of open pores (connected to the outer surface) and closed pores (contrary to open pores) but provide limited information on pore morphology [27]; (ii) SANS cannot provide full-scale porous information on the sample; (iii) Therefore, the pore-skeleton two-phase hypothesis model was chosen to complete the porosity measurement in SANS [40]. However, for samples that are rich in minerals and organic matter, the mineral/mineral-phase scattering is affected, making the selected two-phase model inaccurate for measurements.

### 2.3. Computed Tomography (CT) Scanning

CT scanning utilizes the interaction between X-rays and materials for porous-material characterization or imaging, with the most common scanning mode being based on attenuation scanning for X-rays. Figure 5 shows the CT scan procedure. According to the spatial resolution, these methods can be classified as macro-, micro-, and nano-CT. Macro-CT is used to scan samples with sizes of 10 cm, while the scanning object range of micro-CT is 1 cm–10 μm. Even high-resolution micro-CT cannot capture the entire pore range of 10 nm–10 µm when the pore microstructure is studied [41]. Nano-CT can further characterize the pore microstructure at the submicron scale and compensate for the micro-CT data to a certain extent [23]. However, even the most advanced nano-CT cannot capture the entire sample pore range except for the lower boundary of the capillary pore [42,43].

CT has proven to be an effective method for characterizing the pore morphology and structure of samples [44,45]. The CT method can be applied to study crack evolution after compression, create 3D image tomography, and analyze the sample microstructure [46,47,48,49,50]. Its shortcomings include temporal and spatial-resolution limitations and the problem of distinguishing between material components with similar attenuation coefficients. Moreover, high-resolution applications require a longer display time and a smaller representative sample size, which increases the calculation time, similar to SEM [51].

Figure 6 shows the CT images of Portland cement under different stresses. The aggregate in the CT image is marked in white, while the cracks and pores are marked in black or nearly black. When the stress was up to 36.08 MPa, a crack gradually enlarged. When the axial stress was 12.7 MPa, the specimen entered the damage stage, in which mesocracks propagated and rapidly converged.

### 2.4. Low-Field Nuclear Magnetic Resonance (LF-NMR)

The spin-precession movement of a nucleus exhibits a specific resonance frequency under an external electromagnetic field. When the magnetization vector of the nuclei is disturbed in a direction different from the magnetic field direction, it gradually relaxes towards the latter. The relaxation time of the nuclei depends on the pore structure characteristics of a porous sample because of the interactions between the nuclei and pore surfaces. The magnetization relaxation processes can be polarized and detected by external radio frequency (RF) pulses; thus, microscopic pores are characterized. Compared to expensive solid-state NMR techniques, LF-NMR systems are more suitable for detecting pore-filling fluids (containing protons in the fluid molecules) in many porous materials, and they involve static magnetic fields of the order of a few Tesla and operate at frequencies between 10 and 50 MHz. A limitation of NMR is that the tested sample and fluid must not contain a large concentration of matter, such as ferromagnetic metals and minerals. This significantly affects the external magnetic field.

Figure 7 shows a schematic of the LF-NMR experimental setup for detecting pore-filling methane. The LF-NMR techniques are classified into the following groups: (i) relaxometry; (ii) imaging (NMR imaging, T_1_ or T_2_ spin-echo imaging, or spin-density mapping); and (iii) NMR relaxation tomography, where T_1_ and T_2_ are relaxation times for the longitudinal and transverse directions to the magnetic field, respectively [53]. LF-NMR instruments utilize a shorter spin echo to detect finer pore structures such as micro/nanoscale pore spaces. In a laboratory setting, echoes of 20–100 µs spacing were achieved [54].

Two-dimensional NMR techniques, including T_1_-T_2_ and D-T_2_ (diffusion-T_2_) mapping, have been developed. T_1_-T_2_ mapping is more sensitive to molecular motion in the frequency range between the Larmor frequency (approximately 2 MHz) and extremely low frequencies. Therefore, the T_1_/T_2_ ratio can be used as a parameter to reflect the free and restricted states of the molecules in the fluid [56]. T_1_-T_2_ mapping can also be utilized as a unique probe to distinguish oil-filled pores from organic and inorganic mineral pores, while D-T_2_ mapping can be used to distinguish between oil and water in core samples [54]. However, crystal water in the sample cannot be accurately distinguished because LF-NMR measurements require the sample to be completely saturated with water [57].

Examples of typical LF-NMR results are shown in Figure 8. Figure 8a demonstrates an example of the application of D-T_2_ technique, where the horizontal line represents the diffusion coefficient of water, and the diagonal line is the “oil line,” where the oil signals can be found. The signal is clearly along the oil line, indicating the presence of light oil in the sample. The top and right panels are projections along their respective dimensions [54]. In Figure 8b, the *T*_2_ spectrum is divided into four parts, P1, P2, P3, and P4, which can be defined as (i) the adsorbed methane in micropores, (ii) the porous-medium-confined methane, (iii) the interparticle free methane in the interparticle space of powdered shale, and (iv) bulk methane in the space between shale particles and the inner wall of the sample cell, respectively [2]. The diffusion process of CO_2_ in the n-tetradecane is shown in Figure 8c. In this process, the initial pressure was 5000 kPa and the temperature was 30 °C. It is clear from the image that CO_2_ diffuses gradually towards the bottom in the porous medium, and the concentration eventually tends to be consistent. In addition, the interface gradually increased due to the increase in the volume of the liquid phase when CO_2_ dissolved into the liquid phase.

### 2.5. Summary

Most radiation-detection methods are used to observe the microscopic morphology of a sample, except for LF-NMR, which is, in principle, an indirect measurement method based on fluid saturation. However, unlike other radiation-detection methods, LF-NMR is applicable for large samples, where the sample scale is restricted by the sizes of the RF coils and the auxiliary fluid-saturation system. A core challenge in using LF-NMR for porous-medium characterization is that even though the signals of fluid in nanoscale pores can be detected, identifying the type, phase, and state of the fluid in the pores for the relaxation-time spectra is difficult; this is because the spectra are determined by both the properties of the fluid and the PSD of the material. Different radiation-detection methods are compared in Table 1.

## 3. Fluid-Intrusion Methods

Fluid-intrusion methods require the sample to be fully immersed in gas or liquid, and the measurement is generally conducted in a series of equilibrium states. The core concept is to transfer the measurement of the pore geometry to the quantification of the pore-filling fluid. Fluid-intrusion methods include MIP, gas expansion/intrusion methods, WIP, and adsorption analysis. These methods have a wide application range in porosity measurement, and their measurement results are considered reliable [1,59,60].

### 3.1. Mercury Intrusion Porosimetry (MIP)

MIP is based on the Wasllbum equation and capillary phenomenon, providing two types of measurements: high-pressure and constant-speed MIP [40,61].

High-pressure MIP can enter the pore-throat space (>2 nm) and has a wide range of microscopic pore-structure characterization [1,4]. However, the amount of mercury in the throat and pores cannot be identified separately, and the experimental results are generally greater than the real values in high-pressure mercury porosimetry experiments owing to the pore-space enlargement caused by the high mercury-injection pressure and wetting lag phenomenon [18].

In constant-speed MIP, mercury enters pores at a constant speed, and its experimental state is closer to the real mercury-injection state. The average porosity and permeability measured by constant-speed MIP were lower than those measured by gas, indicating that the pore volume and PSD of the inaccessible part still have a significant influence.

MIP is used for the high-precision measurement of open pores. It is a promising technique for various materials and has several applications [18,26,27,52]. However, it is not suitable for detecting micropores (<2 nm) because mercury does not fill every pore [1,62]. Furthermore, the MIP application is impeded by the following drawbacks: (i) The measurement results of crushed powder are larger than those of plug samples because of the destruction of the pore structure during the crushing process [5,63]; (ii) Small pores are measured under high pressures, which may damage the samples [64].

### 3.2. Gas-Expansion/Intrusion Method

The gas-expansion method is derived from Boyle’s law. The pore volume of a sample is calculated by measuring the pressure change during gas expansion. Helium is used as the medium because it is the smallest nonadsorptive gas molecule and can successfully penetrate the sample’s entire structure [19,65]. The helium-expansion method is used to measure connected pores.

The gas-expansion method is a reliable, well-established, and commercialized measurement method. Various measuring instruments have been developed, such as true volume and density-measurement instruments, porosity-measuring instruments, and pore-size analyzers, with an accuracy of ±0.03%. These instruments use helium, nitrogen, and other inert gases as filling gas. Furthermore, they have a high degree of automation, are easy to operate, and are essential in the field of porosity measurement. Fu et al. measured the total porosity of shale with a volumetric analyzer and a solid densitometer and discussed the factors affecting the measurement results [66]. Sun et al. used a porometer to measure the helium porosity and density of shale samples. The results were compared with those obtained from gas (CO_2_ and N_2_) adsorption and SANS [25]. Zhou et al. used an AP608 automated permeameter-porosimeter to measure the helium porosity and air permeability of coal for comparison with CT and MIP results [45].

Compared with WIP and MIP, the gas-expansion method has a shorter analysis time, simpler operation, and weaker influence on the samples; therefore, it is convenient for repeated measurements [66]. Compared with crushed samples, plug samples underestimate shale porosity [62,67]. In general, this volume-calculation process is based only on the initial and final pressures and has high requirements for temperature control during measurement. Repeated vacuumization and inflation are required for multiple measurements. The opening process of the balance valve inevitably increases the volume of the tube system, which causes systematic measurement errors.

### 3.3. Water-Immersion Porosimetry (WIP)

WIP uses the weight of the sample under dry conditions and the weight difference between air and water after the sample is fully immersed in water to indirectly calculate porosity. This method is suitable for measuring samples with low porosity (<5%) [5]. It is called keroseimmersion porosimetry (KIP) when kerosene is used, and dual-liquid porosimetry (DLP) when both kerosene and water are used [68]. The solutions usually comply with the following conditions [5]: (i) low surface tension and viscosity, and high moisture; (ii) high vapor pressure and low evaporation rate; (iii) does not easily react with samples; (iv) stable composition and density; (v) are harmless and can be disposed of safely.

Tomasz et al. [68] used DLP on shale samples from the Podhale and Baltic Basins to quickly measure clay-bound water (CBW) at 40–80% relative humidity (RH) without crushing. CBW_min_ (40% RH) provides the bound-water value under high hydrocarbon saturation, and CBW_max_ (80% RH) represents the maximum water content of the bound water. The results indicated that WIP is suitable for shale with low density, a high diagenesis degree, and strong cementation, and can be used to calculate hydrocarbon reserves. DLP complements WIP and KIP and obtains the CBW range of rock debris, providing more useful information for formation evaluation.

WIP can keep the samples intact and thereby does not change the sample’s composition [69]. It also has the advantages of low measurement cost, repeatability, and high reliability. However, this method requires the sample to be dried at 200 °C until it reaches a constant weight; this may remove the crystal water in the sample and change the pore structure, causing increased porosity [70]. Simultaneously, the sample must be fully saturated, which takes a long time. Similar to NMR, the clay minerals and organic matter in shale are prone to irreversible chemical reactions with intrusive water phases, damaging the samples and resulting in large measurement results [64].

### 3.4. Adsorption Analysis

Adsorption analysis is based on the capillary aggregation phenomenon and the principle of volume-equivalent substitution. For conventional test fluids such as carbon dioxide and nitrogen, isothermal adsorption tests are usually conducted under low-temperature conditions with the test pressures under the corresponding saturation pressure. Under the assumption that the pore shape is cylindrical and tubular, a capillary-aggregation model was established to estimate the PSD characteristics and pore volume (PV) of the sample [71]. The volumetric method for adsorption was applied by measuring the pressure change caused by capillary aggregation. Figure 9 shows a typical system diagram for nitrogen-adsorption analysis. This volumetric method can effectively characterize the PSD of micropores and mesopores (2–50 nm) in the sample compared to the MIP method [72]. Overall, adsorption-analysis techniques are well-established and widely applied for material characterization.

When adsorption tests are conducted at high pressures and high temperatures (HTHP, for the critical states of the test fluids), the pore-filling fluid usually exists in the gaseous phase; hence, the isotherms cannot provide PSD or PV analysis without capillary aggregation. However, high-pressure and high-temperature adsorption tests show the real state of the fluid in the porous material in application scenarios. In addition to the volumetric method, the gravimetric method, which uses a high-precision balance to measure the gravity change of a sample due to adsorption under various pressures, is also widely applied in HTHP adsorption tests. The gravimetric method is widely used to determine the gas adsorption and sorption capacities of coal and shale [73,74,75,76].

Similar to other fluid-immersion methods, the adsorption analysis also presents the following challenges. (i) Shrinkage and swelling are induced when the fluid is in contact with the porous structure. Most of these effects are irreversible and increase the pore volume, although there are some exceptions—for example, the shrinkage and swelling caused by the contact of CO_2_ and the coal molecular structure are reversible [19]. (ii) Adsorption analysis is a method used to characterize the pores in principle, whose final measurement results are based on a hypothetical pore model [14]. (iii) Accurate nitrogen-adsorption tests require the sample to be dried and vacuumed before the experiment, which may alter the pore structures of some samples [40]. Different fluid-intrusion methods are compared in Table 2.

## 4. Combination of Various Measurement Methods

In recent years, many scholars have combined different porosity-measurement methods to conduct further comparative analyses of various aspects of the samples and clarified the characteristics of different methods and what to consider when selecting methods. Table 3 summarizes part of the measurement work conducted in recent years with respect to sample type and size, measurement method and conditions, and measurement results and conclusions.

### 4.1. Combination of LF-NMR and Other Methods

As introduced previously, the LF-NMR technique, as a noncontact method, can obtain the distribution of hydrogen-containing components in the sample, but the amount of substance identified by NMR is related to the chemical composition of the material and the PSD, which requires special investigation for different materials. In recent years, many studies have focused on various materials, adopting a combination of NMR and fluid-intrusion methods to further understand the NMR measurement results.

Coal and shale are complex natural-gas-storage materials that have been well-studied by combining LF-NMR and other methods. The combination of NMR and adsorption analyses can distinguish the type of methane contained in coal and shale, providing a quantitative amount of methane and some information on methane migration. An NMR method to characterize the adsorption capacity of coal for methane was established by Yao et al., and the measurement results were compared with the HTHP volumetric adsorption-analysis results [55]. They found that the adsorption amount measured via NMR was less than that measured via methane-adsorption (MA) analysis. The explanation was that the methane in coal exists on the pore surface and in solid solutions, which LF-NMR cannot detect [79,80]. In 2018, Yao’s team used NMR and methane/nitrogen adsorption to evaluate and compare the gas content of shale and reached similar conclusions. They discovered that NMR could identify the type of methane gas (free or adsorbed) and quantify free methane in shale. However, MA measurement only quantified the amount of adsorbed methane instead of free gas [2]. In recent years, many scholars have used NMR to monitor the adsorption and gas flow of methane in coal, demonstrating that NMR can provide more detailed methane-flow information [2,81,82].

When NMR is combined with WIP, it can detect free water inside a dry sample, and WIP can determine the mass change before and after water immersion. A recent study by Zhao et al. supports that the measurement results of the two methods are consistent [24]. Before the water-immersion process began, the NMR signals of all samples were not equal to zero, proving that free water still existed in the samples after drying.

NMR provides more pore information than MIP. In the measurement of cement by Zhao et al. ([18]), the MIP data show that there is no pore size smaller than 10 nm, which were the predicted result that refers to the true gel pores or pores generated by pressure damage. By contrast, the NMR method is suitable for the distinction of capillary (<100 nm) and gel pores (200–1000 nm) based on the T_2_ curve, where the first two T_2_ peaks (with an increasing relaxation time) represent the capillary and gel pores, respectively. The PSD of three groups of coal-cutting samples with different particle sizes was measured by Chang et al. using MIP and NMR [26]. The MIP results for cuttings of different particle sizes differed significantly, owing to the influence of interparticle voids and intrusion pressure. NMR is independent of the sample size and shape and thus provides more accurate porosity information than MIP. When the particle size of the cuttings was large (≥1 mm), most of the pores inside the cuttings were intact, providing reasonable porosity-analysis results.

Combining the LF-NMR method with fluid-intrusion methods is an effective approach to understanding the component morphology inside the sample and provides evidence to explain the LF-NMR T_2_ peaks related to the physical properties of the fluid in pores, which improves the reliability and accuracy of LF-NMR measurements. In addition to combining with fluid-intrusion methods, NMR can also be combined with radiation methods, such as SANS, to obtain more information about the sample. Further work in this area is required.

### 4.2. Combination of Different Fluid-Intrusion Methods

Fluid-intrusion methods are well-established measurement methods; however, when any one is used alone, it provides limited pore-size information. The measurement results obtained using different fluid-intrusion methods are not identical for the same sample. Therefore, it is critical to use different fluid-intrusion methods to conduct a comprehensive material analysis.

A more complete PSD range can be obtained by combining different fluid-intrusion methods than only using an individual method. At present, there is a consensus that the molecular size of the medium affects the measurement results. Mercury cannot enter tiny pores owing to the influence of surface tension, and the measurement result is usually the smallest. Sun et al. ([25]) concluded that the helium-expansion method measures open pores with a diameter smaller than 0.2 nm, the CO_2_-adsorption method (CA) measures open pores with a diameter of 0.3–1.4 nm, and the N_2_-adsorption method (NA) quantifies openings with a diameter of 1.4–300 nm. The porosity result of the helium-expansion method was slightly larger than the NA/CA result, owing to the small size of the helium molecules. Similarly, the case reported by Wang et al. ([4]) indicates that the total porosity results obtained via helium expansion are more accurate than those of CA and mercury intrusion. The CA, NA, and MIP data can be used to define the microporous, mesoporous, and macroporous PSDs, respectively. A combination of the three methods can be used to obtain the entire PSD range. Wang et al. ([27]) held the same opinion, considering that CA is suitable for characterizing the porosity and SSA of micropores (<2 nm), NA and SANS are suitable for mesopores (2–50 nm), and MIP is suitable for macropores (>50 nm). Mastalerz et al. ([78]) used NA to calculate the mesoporosity of shale and CA to analyze micropores; their results showed that the SSA obtained via CA was larger than that of NA.

In addition to the measurement medium, which affects the porosity measurement results, the chemical composition of the sample also affects the accuracy of various measurement methods. When CO_2_ and methane molecules come in contact with coal, shrinkage and swelling occur. Rodrigues et al. found that the volume of adsorbed CO_2_ tends to be much higher than the free-gas volume when in contact with the coal structure [19]. The shrinkage and swelling effects of carbon dioxide on the coal structure were completely reversible. Methane also induces shrinkage and swelling when it comes into contact with the molecular structure of the coal. Although these effects are smaller than those of carbon dioxide, they are irreversible and increase the coal volume. In the study by Mastalerz et al., for coal, the SSA measurement result was NA < SANS < CA, and for shale, NA < CA < SANS, which may be due to the expansion of organic matter in the coal sample during the CO_2_-adsorption process [78].

For cementitious materials, intrusion water can react with hydration products when using WIP, increasing the total mass of the intrusion water and improving the measurement results. Recent cases reported by Qian et al. ([14]) indicate that the porosity-measurement results of cementitious materials follow MIP < GIP < WIP.

Inaccessible porosity should be considered when choosing an appropriate porosity measurement method. In the experiment by Wang et al. ([27]), the PSDs obtained from SANS, NA, and MIP seem to be reasonably consistent for most of the tested shale samples. However, for QD_1_-L3, WX_2_-8, and WX_2_-33, the results show that the PSD estimated by SANS is larger than that estimated by MIP in the pore size range of 100–300 nm, indicating the existence of inaccessible porosity.

In summary, when using fluid-intrusion methods, an appropriate measurement medium and method should be selected according to the sample composition. If necessary, various methods can be combined to obtain a complete sample PSD.

## 5. Conclusions

This paper reviews various experimental methods for measuring porosity and porous media. These experimental methods can be classified as radiation-detection and fluid-intrusion methods. Each of these methods can obtain the pore-size information of a sample, but their combined use can provide more accurate information than individually, which is of great significance for understanding the gas-storage and transport mechanisms in porous materials. Based on the review presented herein, the following conclusions are drawn:MIP, the gas-expansion/intrusion method, and adsorption analysis are the most developed measurement technologies. Several developed commercial instruments based on these methods are available. Adsorption analysis mainly focuses on qualitative characterization, and the accuracy of quantitative results for pore volume is questionable. Fluid-intrusion methods can change the sample characteristics to different degrees, among which MIP is the most prominent, and gas-adsorption/intrusion analysis causes marginal damage.SEM and CT are widely used for material analysis and characterization. The quantitative measurement of pore volume requires numerous slices, multi-angle measurement, and 3D reconstruction, and has low measurement efficiency and accuracy. SANS can provide information regarding pore sizes of 1 nm–10 μm but provides limited information on pore morphology. The SEM, CT, and SANS measurement results are limited by the measurement scale.LF-NMR can quantitatively characterize the material itself and the hydrogen-containing fluid in the pores of the material; however, the explanation of the T_2_ spectrum or other two-dimensional spectra needs to be interpreted in combination with other methods. Typically, LF-NMR is used in combination with MIP to clarify the meaning of the T_2_ spectrum.

The pore volume results of a material obtained by different methods are not consistent. Some methods can yield similar results, while others show clear differing trends depending on both the physical mechanism of the method and the properties of the material. Based on comparative studies on the porosity measurement methods listed in Table 1, the following conclusions can be drawn:

For shale and cement, (WIP/NMR/MA/MIP/NA) < CA < SANS < He;

For coal, (MIP/WIP/NMR-NA/SANS) < MA/CA < He.

For the measurement properties of different methods, suitable methods can be selected according to the characteristics of the samples, or multiple methods can be compared or combined to obtain more accurate measurement results. At present, various joint experiments are combined with NMR experiments or adsorption analysis and other methods. However, there is still scope for development.

## Figures and Tables

**Figure 1 materials-15-02981-f001:**
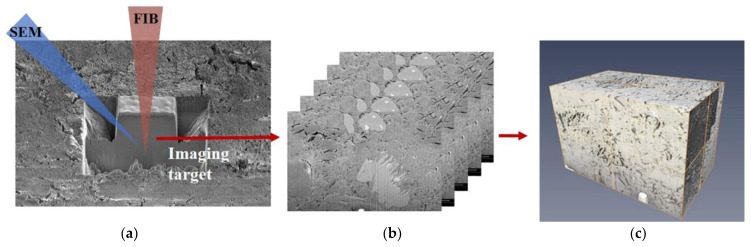
SEM morphology slice-and-view process of cement-paste samples [31]. (**a**) Milling and imaging, (**b**) 2D image-stack registration, (**c**) 3D reconstruction.

**Figure 2 materials-15-02981-f002:**
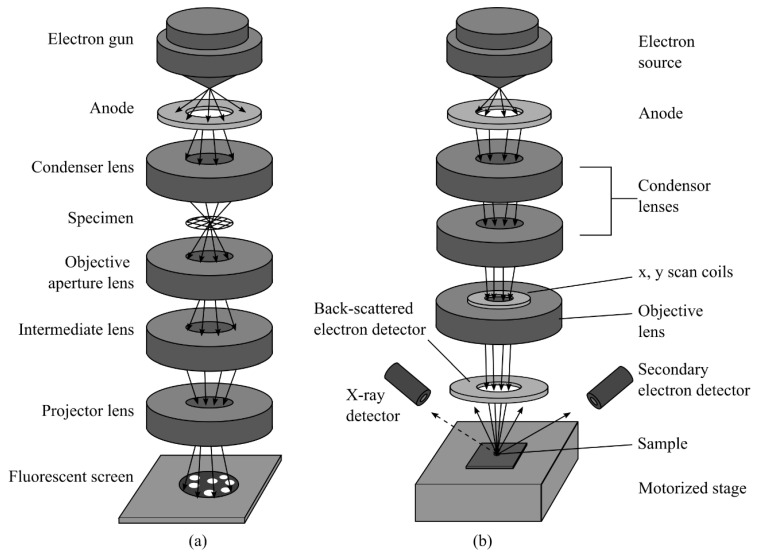
The schematic the optical designs of (**a**) TEM and (**b**) SEM.

**Figure 3 materials-15-02981-f003:**
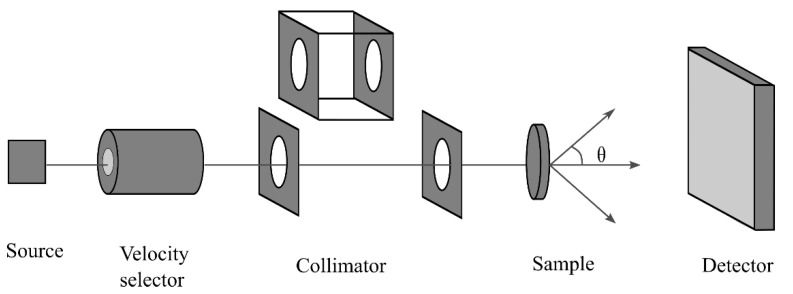
Schematic diagram of SANS (modified from [22]).

**Figure 4 materials-15-02981-f004:**
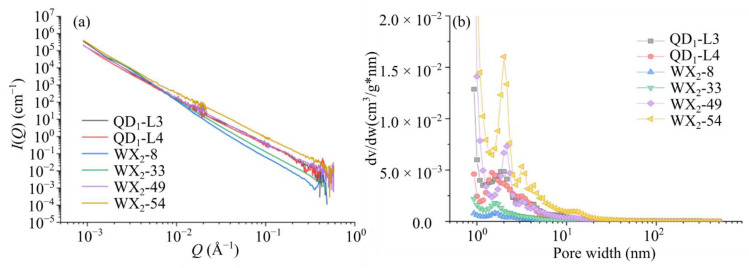
(**a**) Scattering profiles in log−log plots: *I*(*Q*) versus *Q*; (**b**) PSD of shale samples ([27].).

**Figure 5 materials-15-02981-f005:**
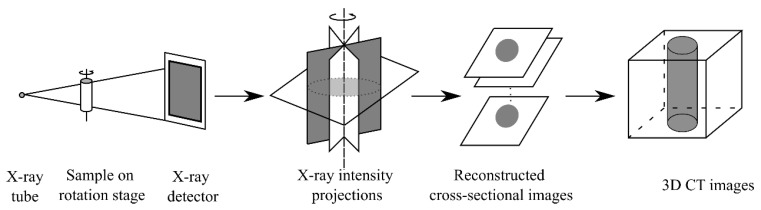
X-ray CT scan procedure (modified from [41]).

**Figure 6 materials-15-02981-f006:**
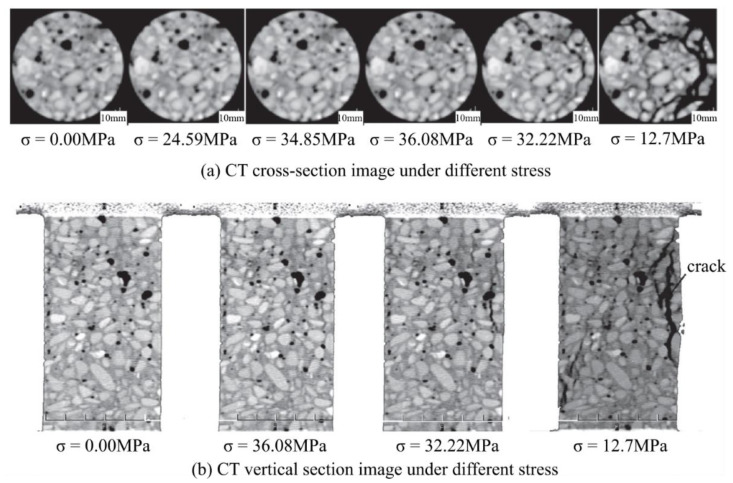
2D CT image of Portland cement under different stresses ([52]).

**Figure 7 materials-15-02981-f007:**
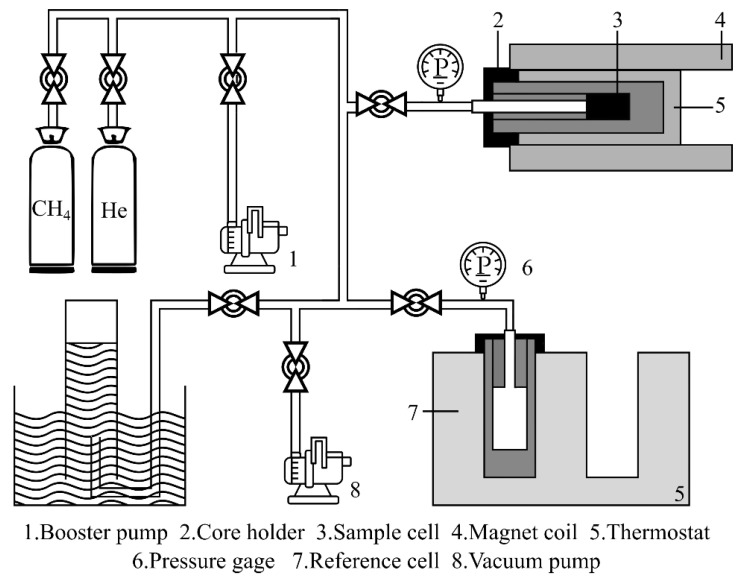
Schematic of the LF-NMR experimental setup (modified from [55]).

**Figure 8 materials-15-02981-f008:**
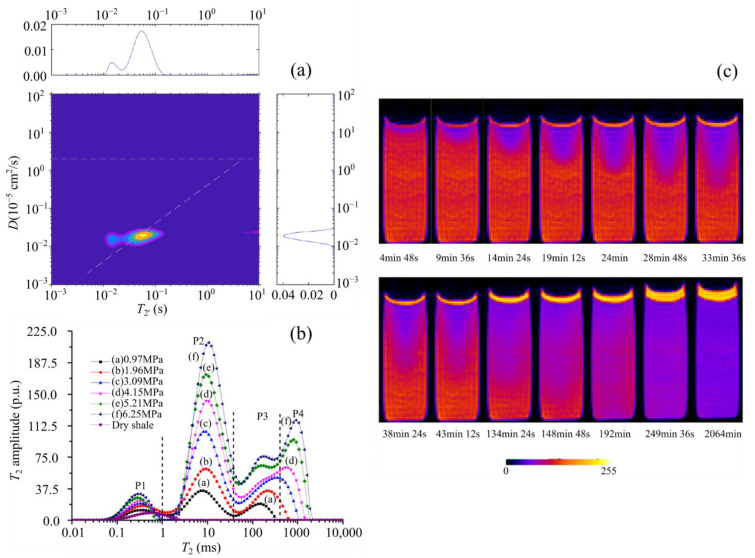
Examples of NMR results: (**a**) D-T_2_ mapping measured at 2 MHz for a shale sample from the Smackover formation (from [54]) (**b**) T_2_ spectra of methane in shale under different pressures (from [2]); the porosity can be calculated according to the calculation of the shaded area. (**c**) 2D NMR images of the CO_2_-diffusion process in porous media saturated with n-tetradecane (from [58]).

**Figure 9 materials-15-02981-f009:**
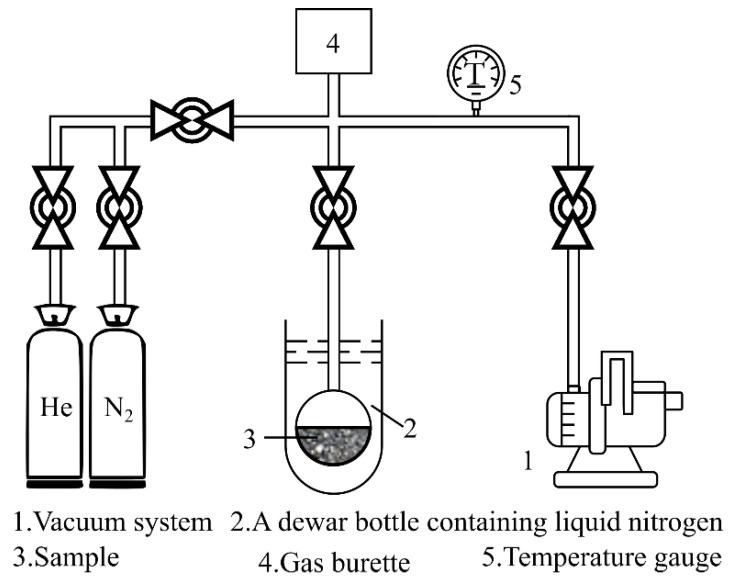
Schematic of the volumetric adsorption method.

**Table 1 materials-15-02981-t001:** Comparisons of different radiation detection methods.

Method	Using Ray	Disadvantages	Advantages
Electron microscopy	Electron beams	Limitation of thin slices; complex image processing	Direct observation of the sample.
SANS	X-rays or neutron rays	Little information on pore morphology.	Characterization of open pores and closed pores.
CT	X-rays	Limitation of temporal resolution and spatial resolution	3D image tomography
LF-NMR	Excitation of hydrogen atoms by electromagnetic radiation	Constraints of identifying the type, phase, and state of the fluid in the pores	Detection of large samples

**Table 2 materials-15-02981-t002:** Comparisons of different fluid-intrusion methods.

Method	Common Measurement Medium	Disadvantages	Advantages
MIP	Mercury	Destruction of the pore structure	Characterization of macropore and mesopore
Gas expansion/intrusion method	He	High requirements for temperature and leakage control; volume increment caused by valve-opening process	Easy operation; repeatability
WIP	Water	Limitation for samples containing clay minerals and organic matter	Low measurement cost; repeatability
Adsorption analysis	N_2_, CH_4_, CO_2_	Shrinkage and swelling caused by CH_4_ and CO_2_	Characterization of micropores and mesopores

**Table 3 materials-15-02981-t003:** Joint-method literature review.

Authors	Sample	Sample Size	Method	Test Condition	Result and Conclusion
Chang et al., 2020 [26]	Coal from the Qinshui and the Junggar basins, China.	1:Plugs:D: 2.5 cm, L: 6 cm 2:Particle: 1.00–1.70 mm; 2.36–3.35 mm; 4.75–6.70 mm	NMR; MIP; Expansion(He)	NMR: 0.53 T magnetic Strength, 23 MHz. MIP:200 MPa	*ϕ*_He_ > *ϕ*_NMR_ The measurement results of NMR and MIP are consistent
Yao et al. 2014 [55]	Coal from southeastern Qinshui Basin, China	Powder with a 60–80 mesh size	NMR; MA	NMR: CPMG sequences of 18,000 echoes, echo spacing 0.3 ms, trains 64. PA:25 °C, 6 MPa	Adsorption capacity: NMR < MA
Yao et al., 2019 [2]	1. Shale from Hunan province in south China. 2. Shale from Sichuan province of southwestern China.	n/a	NMR; MA	NMR: 23.15 MHz, 0.54 T magnetic strength Adsorption: n/a	Adsorption capacity: NMR < MA
Wang et al., 2017 [4]	Shale from Shihui Trough, eastern Qaidam Basin, China.	Ultrafine particle size	NA; CA; MIP; SEM; Expansion(He)	NA/CA: 77 K MIP: 3000 psi Expansion: 28 °C, 1.2 MPa	Porosity < 10 um, mainly mesoporous *ϕ*_He_ > *ϕ*_NA/CA/MIP_; the expansion result is more accurate than other methods
Wang et al., 2020 [27]	1. Shale in northeast Chongqing near the edge of the Sichuan Basin, China. 2. Shale in northeast Yunnan near the southwestern edge of the UYP, China.	Plugs: D: 1 cm, L: 1 cm	MIP; SANS; CA; NA	MIP: Pressures from 0.14 to 413 Mpa CA/NA:77 and 273 K	PSD result: The measurement results of four methods are consistent.
Zhao et al., 2021 [24]	1. White Portland cement (WPC) 2. Standard sand	1.WPC: SSA of 380 m^2^/kg; Density of 3100 kg/m^3^ 2. Standard sand: Particle sizes ranging from 0.08 mm to 2.0 mm	NMR; WIP	NMR:0.5 T magnetic strength, 21.3 MHz frequency	Water-absorption capacity: The results of LF-NMR imaging measurements are consistent with the WIP method
Zhao et al., 2018 [18]	PII 52.5 Portland cement	n/a	MIP; NMR	NMR:0.42 T magnetic strength, 18 MHz frequency	Pore-size range: The measurement range and the order of magnitude are consistent.
Zuena et al., 2019 [57]	Limestone	MIP:1 × 1 × 4 cm^3^ NMR:5 × 10 × 2 cm^3^	MIP; NMR	n/a	There is a good correlation between the quantitative results obtained by MIP and the qualitative ones observed with NMR.
Sun et al., 2017 [25]	Shale from the northwest of Guizhou province, Southwest China	SANS:10 × 10 × 2 mm^3^ Expansion: Plugs: D: 2.5 cm, L: 3 cm Adsorption:40–80 mesh	SANS; Expansion(He); NA; CA	n/a	*ϕ*_NA/CA_ < *ϕ*_He_
Mergia et al., 2010 [77]	A self-sintering carbon mesophase powder based on petroleum residues	Particle size:1μm	Expansion (He); NA; SANS	n/a	*ϕ*_SANS_ and *ϕ*_NA_ are consistent, He expansion is smaller than SANS.
Mastalerz et al., 2012 [78]	1. Shale from New Albany in Indiana. 2. Coal from the Petersburg Formation in America.	n/a	NA; CA; (U)SANS	NA:77.35 K, 101.3 kPa CA:273.1 K SANS: *λ* = 4.8 Å, 0.002 < *Q* < 0.7 Å^−1^ USANS:*λ* = 2.4 Å, 5 × 10^−5^ < *Q* < 0.003 Å^−1^	SSA result: For coal, NA < SANS < CA For shale, NA < CA < SANS
Shi et al., 2020 [46]	Coal from the Qinshui Basin, China	Plugs: H: 5 cm, L: 2 cm	CT; MIP	n/a	n/a
Qian et al., 2021 [14]	Cement paste	n/a	Expansion(N_2_); MIP; WIP	n/a	*ϕ*_MIP_ < *ϕ*_N2_ < *ϕ*_WIP_
Liu et al., 2019 [40]	Shale	n/a	SANS; NA; MIP	SANS: *λ* = 0.53 nm (Δ*λ*/*λ* = 18%) NA: 77 K, *P*/*P*_0_ range in 0.01–0.99 MIP: injection pressure 0–60,000 psia	SSA result: NA < SANS < MIP

CA: CO_2_-adsorption method. NA: N_2_-adsorption method. MA: CH_4_-adsorption method. SSA: Specific surface area. PSD: Pore-size distribution.

## Data Availability

Not applicable.

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
