# Peer review of "Recent Progress in Single and Combined Porosity-Evaluation Techniques for Porous Materials"

_materials, 2022, doi:10.3390/ma15092981_

Round 1

Reviewer 1 Report

The manuscript “Recent Progress in Single and Combined Porosity Evaluation Techniques for Porous Materials” provides an overview of radiation detection methods and fluid intrusion methods used for characterizing porous materials. Some portions of the manuscript provide a useful review of the literature, however, there are a number of issues with the paper preventing it from being publishable in its current form. Comments are provided below:   

  • The first sentence in the abstract is long and written poorly and should be revised.
  • Some sentences are written in such a way that they seem to be stating something the authors did not intend to state.
    • For example, in the Electron Microscopy Section, how does the resolution allow for observing samples that are 100 – 500 nm thick?
    • And the sentence “TEM should be used in conjunction with other methods to evaluate organic and carbon samples” perhaps is meant to say “TEM should be used in conjunction with other methods when characterizing organic and carbon samples.”
  • What the cement paste samples are in Figure 1 should be described.
  • The acronyms DPSO and HSO are not defined.
  • The first sentence under Figure 2 about high sensitivity of neutrons to hydrogen should be explained. Perhaps giving an example of the sample being considered would help.
  • All symbols in the caption of Figure 3 should be defined. Also, these equations should be explained in the text and should not be in the Figure caption. The samples in the legend are not defined. If Figure 3 is to be used as an example then it should be explained thoroughly and clearly.
  • In general the Figures in the manuscript are not explained. They are just taken from other references and inserted into this manuscript without discussing what they are about. In many cases there are symbols, legends, labels, and text in these inserted images that is not explained and there is no way for the reader to know what these are.
  • The beginning of the second sentence in Section 4 is missing.

Reviewer 2 Report

The present review introduces currently used porosity measurement methods. It provides resources for new researchers in related fields to gain insight into the existing experimental methods and help them select suitable experimental methods. Overall review is interesting, but it needs some major revision.

  1. Author should revise the manuscript thoroughly as it contains many grammatical and syntax errors.
  2. The schematic figure for comparison of all techniques should be provided by considering some points, such as

How these methods are differentiated from each other?

Where we can use the particular method?

What are the constraints and benefits of each method?

  1. Author should provide some insights on use of the particular porosity measurement method with some examples (materials where we can use it) and references.

Reviewer 3 Report

Comments from Reviewer
Title: Recent progress in single and combined porosity evaluation techniques for porous materials
The paper presents research on the recent progress in the accurate determination of porosity and specific surface area of porous materials such as shale and cement. The current form's presentation of methods and scientific results is unsatisfactory for publication in the Materials journal. The minor and significant drawbacks to be addressed can be specified as follows:
1.    The authors must supply an ORCID ID for all authors. Getting an ORCID iD is FREE, quick and easy to do through the ORCID registration page: https://orcid.org/register. Please, give the respective ORCID ID in the manuscript.
2.    Keywords. Why is NMR listed as the only technique, although other ones are discussed in this paper? Please mention others.
3.    Some abbreviations are explained several times in the text. It is sufficient to do it the only first time. For example, see page 2 ("computed tomography (CT) scanning[21-24].") and ("computed tomography (CT) scanning, small-angle").
4.    Page 3. "Error! Reference source not found."?
5.    Fig. 1, figure captions. S. Lim et al.[31] ---> Lim et al.[31]. Please omit the initials of the names. All work should be checked.
6.    Fig. 1. The resolution of images?
7.    Fig. 3, figure captions, "PSD can be obtained". PSD? This abbreviation is explained on Page 8.
8.    Fig. 3. Please give in this figure the respective pore-size distribution.
9.    Schematic diagram of TEM/SEM?
10.    Fig. 9. This figure should be rejected. Better to give some examples of real adsorption isotherms.
11.    Page 10. summarizes part ---> Summarizes part
12.    Page 11. Water absorption? Or adsorption?
13.    Tab. 1. For me, this table is chaotic.
14.    Literature should also be standardized: the size of letters in the titles of journals, initials of names, the size of letters in the titles of articles, abbreviations.    

Sincerely,
The reviewer.

Round 2

Reviewer 2 Report

The revised manuscript can be accpeted.

Author Response

Dear reviewer:

Thank you again for your careful review work.

Yours sincerely

Yuqing Wang

Reviewer 3 Report

The authors have made the essential corrections, provided some detailed answers to some of the questions. Overall the manuscript improved.

After reviewing the revised work, the following remarks arise

  1. Poor quality of figures, i.e. Figs. 4, 6, and 8.
  2. Hardly legible figure, i.e. Fig. 8.

Before the final approval, the authors should be forced to improve these plots/schemes.
